# Prebiotic Combinations Effects on the Colonization of Staphylococcal Skin Strains

**DOI:** 10.3390/microorganisms9010037

**Published:** 2020-12-24

**Authors:** Silvia Di Lodovico, Franco Gasparri, Emanuela Di Campli, Paola Di Fermo, Simonetta D’Ercole, Luigina Cellini, Mara Di Giulio

**Affiliations:** 1Department of Pharmacy, University “G. d’Annunzio”, Chieti-Pescara, Via dei Vestini, 31, 66100 Chieti (CH), Italy; silvia.dilodovico@unich.it (S.D.L.); e.dicampli@unich.it (E.D.C.); paola.difermo@unich.it (P.D.F.); m.digiulio@unich.it (M.D.G.); 2Department of Pharmacy, University of Salerno, Via Giovanni Paolo II, 132, 84084 Fisciano (SA), Italy; info@gasparrifranco.it; 3Department of Medical, Oral and Biotechnological Sciences, University “G. d’Annunzio”, Chieti-Pescara, Via dei Vestini, 31, 66100 Chieti (CH), Italy; simonetta.dercole@unich.it

**Keywords:** skin microbiota, *S. aureus*, *S. epidermidis*, prebiotic combinations, xylitol, oligosaccharides, species-specific action, antimicrobial/antibiofilm effect

## Abstract

Background: An unbalanced skin microbiota due to an increase in pathogenic vs. commensal bacteria can be efficiently tackled by using prebiotics. The aim of this work was to identify novel prebiotic combinations by exerting species-specific action between *S. aureus* and *S. epidermidis* strains. Methods: First, the antimicrobial/antibiofilm effect of Xylitol-XYL and Galacto-OligoSaccharides–GOS combined with each other at different concentrations (1, 2.5, 5%) against *S. aureus* and *S. epidermidis* clinical strains was evaluated in time. Second, the most species-specific concentration was used to combine XYL with Fructo-OligoSaccharides–FOS, IsoMalto-Oligosaccharides–IMO, ArabinoGaLactan–LAG, inulin, dextran. Experiments were performed by OD_600_ detection, biomass quantification and LIVE/DEAD staining. Results: 1% XYL + 1% GOS showed the best species-specific action with an immediate antibacterial/antibiofilm action against *S. aureus* strains (up to 34.54% ± 5.35/64.68% ± 4.77) without a relevant effect on *S. epidermidis*. Among the other prebiotic formulations, 1% XYL plus 1% FOS (up to 49.17% ± 21.46/37.59% ± 6.34) or 1% IMO (up to 41.28% ± 4.88/36.70% ± 10.03) or 1% LAG (up to 38.21% ± 5.31/83.06% ± 5.11) showed antimicrobial/antibiofilm effects similar to 1% XYL+1% GOS. For all tested formulations, a prevalent bacteriostatic effect in the planktonic phase and a general reduction of *S. aureus* biofilm formation without loss of viability were recorded. Conclusion: The combinations of 1% XYL with 1% GOS or 1% FOS or 1% IMO or 1% LAG may help to control the balance of skin microbiota, representing good candidates for topic formulations.

## 1. Introduction

The skin is the largest organ of the human body and plays the pivotal role of protecting the host from pathogenic infections and penetration of harmful agents. The skin is colonized by numerous bacterial species, namely skin microbiota that exerts an important role in the maintenance of cutaneous homeostasis [1]. The composition of skin microbiota varies among individuals and depends on skin topography [2]. A healthy balanced microbiota is a microbial shield against pathogenic microorganisms, it prevents dry skin conditions, improves skin health and modulates host immunity [3].

The most commonly isolated bacterium in a healthy skin microbiota is *Staphylococcus epidermidis*, making up to 90% [4] of the aerobic species. This microorganism balances the inflammatory response after host injury by producing antimicrobial molecules that selectively inhibit skin pathogens such as *Staphylococcus aureus* [5]. *S. aureus* causes blisters, abscesses, redness and swelling of the skin, and it is able to produce a variety of virulence factors like toxins and hemolysins that appear to amplify the severity of skin pathologies such as Atopic Dermatitis (AD). In the end, *S. aureus* is able to form a biofilm that is involved both in the bacterial skin adhesion and in the resistance/tolerance phenomenon against antimicrobial agents. A recent review underlines that to maintain robust cutaneous homeostasis, it is fundamental to limit the skin carriage and invasion of the dermato-pathogen *S. aureus,* preserving the resident microbial population [6]. In fact, if the *S. aureus* colonization increases, *S. epidermidis* and other coagulase-negative staphylococci (CoNS) colonization decreases, resulting in a dysbiosis with unbalanced skin microbiota and tissue damage [6,7,8]. This effect can also be caused by frequent washing, physical stress, topical treatments with antibiotics or detergents, but also due to AD or diabetes [7,8,9]. This imbalance between skin pathogens and commensal bacteria can be tackled by using prebiotics [10]. These compounds are non-digestible (by humans) fermentable food ingredients that promote the growth of beneficial microorganisms, which have the potential to improve host health [10,11,12]. In particular, Galacto-OligoSaccharides (GOS), Lactulose, Polydextrose, Inulin, Fructo-OligoSaccharides (FOS) are classified as established prebiotics, and Xylitol (XYL), IsoMalto-Oligosaccharides (IMO), ArabinoGaLactan (LAG) and dextran are recognized as emerging prebiotics [11].

Among skin emerging prebiotics, noted is XYL (CH_2_OH(CHOH)_3_CH_2_OH), a sugar alcohol used as a sweetener and naturally found in low concentrations in the fibers of many fruits and vegetables, that can be extracted from various berries, oats, and mushrooms, as well as from fibrous material such as corn husks and sugar cane bagasse [12,13,14].

Galacto-OligoSaccharides, also known as oligogalactosyllactose, oligogalactose, oligolactose or trans-galacto-oligosaccharides (TOS) [15,16], occur in commercially available products such as food for both infants and adults. The composition of the GOS fraction varies in chain length and type of linkage between the monomer units. Galacto-OligoSaccharides are produced through the enzymatic conversion of lactose or from botanical sources such as pea fibers.

Previous studies demonstrated that XYL and GOS used alone inhibited glycocalyx production by *S. aureus* cells [12,15,16]. Xylitol seems to be the most versatile and investigated sugar alcohol for its numerous healthy effects on digestive, respiratory tracts, bone, immune function and recently, also on the skin. Xylitol shows the ability to improve the skin barrier function, reducing skin moisture loss and inhibiting the growth of skin pathogenic microorganisms [13].

Based on this data, the dual purpose of this study was: first, to define the antimicrobial and antibiofilm selective effect deriving from XYL and GOS combinations at different concentrations against *S. aureus* and *S. epidermidis* clinical strains; second, the most species-specific concentration detected above was used to screen XYL with novel skin prebiotics such as FOS, IMO, LAG, inulin, dextran.

The goal of this study was to research new skin emerging prebiotic combinations capable of exerting a staphylococcal selective species-specific action while keeping balance in the normal skin microbiota.

## 2. Materials and Methods

### 2.1. Prebiotic Compounds

Xylitol, GOS, FOS, IMO, LAG, inulin, dextran were purchased by Lonza Group Ltd. (Basel, Switzerland).

Each prebiotic compound was prepared in phosphate-buffered saline (PBS, Thermo Fisher Scientific, Milan, Italy) and stored at −20 °C until use. The stock dilutions of XYL and GOS were diluted in PBS to obtain the final concentrations of 1, 2.5, 5% and the stock dilutions of FOS, IMO, LAG, inulin, dextran were diluted in PBS to obtain the final concentrations of 1%.

### 2.2. Bacterial Strains

Anonymized *Staphylococcus aureus* 815, *S. aureus* PECHA 10, *S. epidermidis* 317 and *S. epidermidis* MDG1 clinical strains, coming from the private collection of the Bacteriology Laboratory of Pharmacy Department, were used in this study. *Staphylococcus aureus* 815 [17,18,19], *S. aureus* PECHA 10 [20] and *S. epidermidis* 317 [21] were previously characterized for their biofilm-related properties.

The characterized strains, stored at −80 °C, were cultured in Tryptic Soy Broth (TSB, Oxoid, Milan, Italy) at 37 °C overnight under aerobic conditions. After incubation, the broth cultures were diluted 1:10 in the same medium and refreshed for 2 h at 37 °C in thermostatic shaking (160 rpm) bath. Finally, the cultures were adjusted in a spectrophotometer (Eppendorf, Milan, Italy) to an Optical Density at 600 nm (OD_600_) = 0.12, corresponding to 0.5 Mcfarland [17]. These standardized broth cultures were used for the experiments.

### 2.3. Effect of Prebiotic Combinations on Planktonic Bacteria

The antimicrobial effect of XYL and GOS alone and combined with each other at different concentrations (1, 2.5, 5%) was assessed against *S. aureus* 815, *S. aureus* PECHA 10, *S. epidermidis* 317 and *S. epidermidis* MDG1. Fifty microliters of XYL, 50 μL of GOS (mixed at the above-mentioned concentrations) and 100 μL of each standardized broth culture were inoculated in 96-well microtiter plates and incubated for 3, 6 and 24 h at 37 °C in aerobic condition. The scheme in Figure 1 shows the detected XYL and GOS combinations.

The OD_600_ was measured with an ELISA reader (SAFAS, Munich, Germany) after 3, 6 and 24 h of incubation at 37 °C. The antibacterial effect of XYL and GOS alone and combined with each other was determined as a percentage of OD_600_ reduction in respect to the control during the time.

The chosen XYL and GOS (1% + 1%) most performing combination in terms of selective species-specific antibacterial action and low concentration was used to evaluate the effect of XYL with FOS or IMO or LAG or inulin or dextran (skin emerging prebiotics) against *S. aureus* 815, *S. aureus* PECHA 10, *S. epidermidis* 317 and *S. epidermidis* MDG1, in planktonic phase. Fifty microliters of XYL (at 1%), 50 μL of other prebiotics (at 1%) and 100 μL of each standardized broth cultures were inoculated in 96-well microtiter plates and incubated for 3 and 24 h at 37 °C in aerobic condition.

The percentage of OD_600_ reduction in respect to the controls was analyzed, as described above.

In these experiments, the evaluation of data after 6 h was undetected since the previous 3 and 6 h OD_600_ results related to XYL and GOS did not show significant differences. The detections of OD_600_ reduction after 3 and 24 h were representative of immediate and stable effect, respectively. For the controls, 1% XYL and 1% of FOS, IMO, LAG, inulin, dextran was also assayed alone.

The inhibition percentage of planktonic growth was calculated using the following formula:Inhibition percentage (%) = [(OD_control_ − OD_sample_)/OD_control_] × 100.(1)

The amount of planktonic inhibition was calculated by evaluating the amount of planktonic growth in the absence of prebiotic combinations (defined as 100% planktonic growth) and the media sterility control (defined as 0% planktonic growth).

For each determination, TSB without bacteria (blank) added to the different concentrations of prebiotics alone or combined with each other was used as a negative control.

Each determination was performed in duplicate for five independent experiments.

The most performing prebiotic combinations were also studied for their species-specific action when *S. aureus* PECHA 10 and *S. epidermidis* MDG1 were co-cultured.

Briefly, mixed inocula containing equal CFU/mL (≅10^6^ CFU/mL) of *S. aureus* PECHA 10 and *S. epidermidis* MDG1 were treated with 1% XYL plus 1% GOS or FOS or IMO or LAG and compared to untreated samples (with PBS) after 3 and 24 h of incubation at 37 °C. The CFU/mL enumeration of *S. aureus* and *S. epidermidis* cells was detected after the spreading of 100 μL of serial dilutions of broth cultures on Mannitol Salt Agar (MSA, Oxoid, Milan, Italy). Experiments were performed in duplicate in three independent experiments.

### 2.4. Effect of Prebiotic Combinations on Biofilm

The antibiofilm activity of the different combinations of XYL and GOS at concentrations of 1, 2.5, 5% was performed on *S. aureus* 815, *S. aureus* PECHA 10, *S. epidermidis* 317 and *S. epidermidis* MDG1 mature and in formation biofilms by biomass quantification [22]. For mature biofilm, 200 μL of standardized bacterial suspensions, grown in TSB supplemented with 0.5% (*v*/*v*) glucose, were inoculated on 96-well flat-bottomed microtiter plates and incubated for 24 h at 37 °C. After incubation, the planktonic cells were gently removed, and the wells were washed with sterile PBS, filled with each different combination of XYL and GOS (concentrations: 1, 2.5, 5%) (according to the scheme of Figure 1) and incubated for 5 min and 1 h at 37 °C. After incubation, each well of the 96-well flat-bottomed microtiter plates was washed twice with sterile PBS, air-dried, stained for 1 min with 0.1% safranin and washed with PBS. The stained biofilms were resuspended in 200 μL ethanol (95% *v*/*v*) and measured at OD_492_ by spectrophotometry using an ELISA reader [17].

For *S. aureus* 815 *S. aureus* PECHA 10, *S. epidermidis* 317 and *S. epidermidis* MDG1 biofilms formation, 100 μL of XYL and GOS alone or combined with each other (50 μL of each substance) at different concentrations (1, 2.5, 5%) and 100 μL of standardized broth cultures were inoculated in 96-well flat-bottomed microtiter plates (according to the scheme of Figure 1) and incubated for 3 and 24 h at 37 °C. The evaluation after 3 h of incubation was considered the first step of bacterial adhesion.

After incubation, the produced biomass was quantified, as described above [22].

The inhibition percentage of biofilm formation was calculated using the following formula:Inhibition percentage (%) = [(OD_control_ − OD_sample_)/OD_control_] × 100.

The amount of biofilm inhibition was calculated by evaluating the amount of biofilm that was grown in the absence of prebiotic combinations (defined as 100% biofilm) and the media sterility control (defined as 0% biofilm).

For each determination, TSB with 0.5% (*v/v*) glucose without bacteria (blank) added to the different concentrations of prebiotics alone or combined with each other was used as a negative control.

Similar to the planktonic bacterial population, the antibiofilm activity of 1% XYL combined with 1% FOS, IMO, LAG, inulin, dextran was also performed on *S. aureus* 815, *S. aureus* PECHA 10, *S. epidermidis* 317 and *S. epidermidis* MDG1 biofilms formation. Briefly, 100 μL of each prebiotics combination (50 μL 1% XYL + 50 μL 1% FOS or IMO or LAG or inulin or dextran) and 100 μL of standardized broth cultures were inoculated in 96-well flat-bottomed microtiter plates and incubated for 3 and 24 h at 37 °C. After incubation, the produced biomass was quantified as described above. The antibiofilm action of 1% XYL plus 1% FOS or IMO or LAG or inulin or dextran was compared with the antibiofilm action obtained with 1% XYL plus 1% GOS that was considered the most performing combination in terms of selective species-specific antibiofilm action against Staphylococci in the previous tests.

Each determination was performed in duplicate for four independent experiments.

### 2.5. Viability Test and Microscopic Observations

Bacterial viability in planktonic and sessile phases treated with XYL and GOS alone and combined with each other at each studied concentration and with 1% XYL plus 1% FOS, IMO, LAG, inulin, dextran were examined with the LIVE/DEAD staining (Molecular Probes Inc., Invitrogen, San Giuliano Milanese, Italy) as indicated by the manufacturer and visualized under a fluorescence Leica 4000 DM microscope (Leica Microsystems, Milan, Italy). Briefly, for the planktonic phase, after 24 h of treatment at 37 °C, 20 μL of each tested condition was centrifuged, and the pellets were resuspended and stained with LIVE/DEAD staining for 15 min in the dark and examined for its viability. For the sessile phase (biofilm formation), after 24 h of treatment at 37 °C, the supernatant was removed by aspiration and each well, washed twice with PBS, stained with LIVE/DEAD kit for 15 min in the dark, washed once again and examined for the sessile population viability.

LIVE/DEAD analysis was performed to qualitatively evaluate the prebiotics effect on tested strains. Bacteria stained in red (propidium iodide) expressed compromised membrane integrity (damaged), whereas green stained bacteria (SYTO 9) remained viable. The images were recorded at excitation/emission wavelengths of 485/498 nm for SYTO 9 and 535/617 nm for propidium iodide. The analysis of viable cells in the planktonic phase and viable and clustered cells in the sessile phase was determined by examining ten random fields of view for each sample, and each slide was controlled by three blinded microbiologists [20,23,24].

### 2.6. Statistical Analysis

All data were obtained from at least three independent experiments performed in duplicate. Data were shown as the means ± standard deviation (SD). The impact of each prebiotic combination on each species growth or biofilm formation ability was evaluated in respect to the control (PBS) or reference control (% XYL + 1% GOS for skin emerging prebiotic combinations) by using one-way analysis of variance (ANOVA). *p* values ≤ 0.05 were considered statistically significant.

## 3. Results

The antimicrobial activity of XYL and GOS alone and combined with each other was evaluated. Table 1 shows the percentages of bacterial growth reduction in respect to the controls in presence of all tested combinations of XYL and GOS. For *S. aureus* 815, interesting percentages of planktonic growth reduction were obtained with all tested combinations with a major effect after 3 and 6 h that decreased in time (24 h). The percentages of planktonic reduction ranged from 13.80% ± 13.79 (0% XYL + 1% GOS) to 40.60% ± 15.77 (5% XYL + 5% GOS) at 3 h, from 15.86% ± 7.17 (1% XYL + 0% GOS) to 55.50% ± 8.78 (5% XYL + 5% GOS) at 6 h and from 5.47% ± 3.62 (1% XYL + 0% GOS) to 22.53% ± 7.27 (2.5% XYL + 5% GOS).

A similar effect was detected for *S. aureus* PECHA 10 with the major effect at 3 h in all tested conditions. The percentages of planktonic reduction ranged from 4.07% ± 5.52 (0% XYL + 1% GOS) to 62.29% ± 2.95 (5% XYL + 5% GOS) at 3 h, from 1.21% ± 1.71 (0% XYL + 1% GOS) to 44.83% ± 1.98 (2.5% XYL + 5% GOS) at 6 h and from 0.00% ± 0.00 (0% XYL + 1% GOS) to 18.46% ± 3.72 (2.5% XYL + 5% GOS).

Regarding *S. epidermidis* strains, a general lower effect than *S. aureus* strains was produced after XYL and GOS treatment.

From the analysis of data, the combination 1% XYL plus 1% GOS showed the most effective in terms of species-specific inhibition rate. This combination displayed a general higher activity against *S. aureus* 815 and *S. aureus* PECHA 10 than *S. epidermidis* strains, especially in respect to the effect detected in *S. epidermidis* MDG1 at 3 h. The species-specific action of XYL and GOS against *S. aureus* and *S. epidermidis* strains was detected in all tested times. In particular, with 1% XYL and 1% GOS, a relevant planktonic reduction was obtained with *S. aureus* 815 and *S. aureus* PECHA 10 strains in all detected times up to 34.54% ± 5.35 at 6 h for *S. aureus* 815 and 27.08% ± 9.21 at 3 h for *S. aureus* PECHA 10 without significant action on *S. epidermidis* growth (1.33% ± 1.89 and 6.43% ± 10.98 at 3 and 6 h for *S. epidermidis* MDG1).

Hence, 1% XYL and 1% GOS was chosen as the most selective species-specific combination in terms of low concentration/good action for all analyzed strains.

In Appendix A reported are the mean values of OD_600_ for the planktonic phase obtained combining all concentrations of XYL and GOS against the detected strains.

The antibiofilm effect of XYL and GOS alone and combined with each other was performed by biomass quantification. No relevant percentage of mature biofilm reduction was detected for each combination after 5 min and 1 h with respect to the controls (data not shown).

Figure 2 shows the percentages of reduction of *S. aureus* 815, *S. aureus* PECHA 10, *S. epidermidis* 317 and *S. epidermidis* MDG1 biofilm formation after 3 and 24 h. It was notable a marked reduction of bacterial adhesion up to 39.60% ± 8.27 and 82.95% ± 8.93 after 3 h of treatment up to 51.53% ± 5.04 and 96.56% ± 4.87 after 24 h for *S. aureus* 815 and *S. aureus* PECHA 10, respectively. In respect to the *S. aureus* strains, the tested combinations of GOS and XYL expressed low reduction values of *S. epidermidis* adhesion and biofilm formation at the same time. In fact, the best percentages of adhesion and biofilm reduction were 19.77% ± 4.66 and 22.18% ± 3.98 after 3 and 24 h for *S. epidermidis* 317 and 62.02% ± 5.36 and 57.76% ± 4.80 after 3 and 24 h for *S. epidermidis* MDG1. In particular, 1% XYL + 1% GOS displayed a selective species-specific effect for the antibiofilm action with 29.44% ± 10.97 for *S. aureus* 815 and 64.68% ± 4.77 for *S. aureus* PECHA 10 vs. 2.50% ± 0.71 for *S. epidermidis* 317 and 2.39% ± 3.38 for *S. epidermidis* MDG1 after 3 h; 35.91% ± 16.62 for *S. aureus* 815 and 41.64% ± 2.99 for *S. aureus* PECHA 10 vs. 8.34% ± 6.60 for *S. epidermidis* 317 and 0.00% ± 0.00 for *S. epidermidis* MDG1 after 24 h.

The mean values of produced biofilm biomasses obtained with the different combinations of XYL and GOS for all detected strains are shown in Appendix A.

For all tested strains, 1% XYL and 1% GOS also represented the best selective species-specific antibiofilm combination in terms of low concentration/significant antibiofilm action, especially at 3 h.

Based on these results, the combination 1% XYL plus 1% GOS was chosen as the reference control (RC) to screen XYL combined with 1% FOS or IMO or LAG or inulin or dextran both for antimicrobial and antibiofilm evaluations.

As shown in Table 2, except for 1% XYL plus 1% dextran (at 3 h) and plus 1% IMO or LAG or inulin (at 24 h), good percentages of *S. aureus* 815 planktonic growth reduction were obtained with all tested combinations after 3 and 24 h. The major percentages of *S. aureus* 815 growth reduction were obtained when XYL was combined with FOS (49.17% ± 21.46) and with dextran (44.26% ± 1.01). Regarding *S. aureus* PECHA 10, interesting percentages of planktonic reduction in respect to the control were obtained in all tested combinations at 3 h with the best effect when XYL was combined with LAG (38.21% ± 5.31). The planktonic effect at 24 h was less than 3 h, with the major effect obtained with the reference control (11.22 ± 15.86). As regards *S. epidermidis* 317, the planktonic growth reduction was obtained after 3 h with XYL, and FOS (percentage of reduction: 24.94% ± 10.82) and XYL and inulin (percentage of reduction: 26.27% ± 11.57) did not display a selective species-specific action against the studied microorganisms. The combinations XYL with IMO and XYL with LAG showed a selective effect after 3 h; meanwhile, the combination of XYL with dextran displayed a species-specific action after 24 h. For *S. epidermidis* MDG1, the major species-specific action was obtained with XYL + FOS at 24 h. Xylitol combined with LAG and IMO at 3 h for all detected strains showed an antimicrobial species-specific action also comparing with the reference control. At 24 h, a species-specific action was obtained only comparing *S. aureus* 815 and *S. epidermidis* strains with 1% XYL + 1% FOS.

Appendix A shows the OD_600_ obtained combining 1% XYL and 1% other prebiotics against all tested strains.

All tested combinations showed a good antibiofilm effect at both tested times with a major effect after 24 h against *S. aureus* PECHA 10 in respect to the reference control (Figure 3). In particular, the best effect, in terms of percentage of *S. aureus* 815 adhesion and biofilm reduction, was obtained with XYL with LAG (29.40 ± 8.33 at 3 h and 55.73% ± 16.74 at 24 h). Regarding *S. aureus* PECHA 10, a remarkable antiadhesive/antibiofilm effect was detected with percentages of biofilm reduction up to 100.00% ± 0.00 at 3 h with XYL + FOS and up to 90.32 ± 3.77 at 24 h with XYL + FOS. Regarding *S. epidermidis* 317, except for the combinations XYL with FOS (3 h), XYL with LAG (24 h) and XYL with dextran (24 h), all tested combinations showed a poor ability to inhibit the adhesion and the biofilm formation at 3 and 24 h. At 3 h, XYL combined with IMO and LAG displayed low percentages of adhesion reduction against *S. epidermidis* MDG1. At 24 h, interesting percentages of biofilm reductions were determined with values ranged from 0.00% ± 0.00 (with XYL + GOS) to 58.36% ± 8.34 (with XYL + IMO).

The combination XYL with dextran did not show a selective species-specific antibiofilm effect at the two detected times against the studied microorganisms.

Xylitol combined with IMO or LAG (at 3 and 24 h) showed a selective species-specific effect on the sessile population of the studied microorganisms.

In Appendix A, the mean values of produced biomass in the presence of XYL and other skin emerging prebiotics for all detected strains are shown.

The best-performing prebiotics combinations (1% XYL + 1% of GOS or FOS or IMO or LAG) also displayed a species-specific effect when Staphylococcal strains were co-cultured (Figure 4). After 3 h, in the control sample (condition with PBS), a significant prevalence (*p* < 0.05) in *S. aureus* PECHA 10 growth was detected; on the contrary, in the presence of prebiotics combinations, the *S. aureus* PECHA 10 values were lower than the *S. epidermidis* MDG1 values with significance for 1% XYL combined with 1% IMO or LAG (*p* < 0.05). In all prebiotic combinations, a significant decrease in the *S. aureus* PECHA 10 CFU/mL was observed; whereas, the *S. epidermidis* MDG1 growth was not significantly reduced for the conditions 1% XYL + 1% GOS or 1% FOS and was significantly overexpressed for the conditions 1% XYL + 1% IMO or 1% LAG in respect to the control sample.

After 24 h, the species-specific action was less detected with a balanced effect after treatment with 1% XYL plus 1% FOS or 1% IMO. A significant *S. aureus* PECHA 10 decrease in the presence of 1% XYL + 1% IMO in respect to the control was detected; whereas, in all tested prebiotic combinations, the *S. epidermidis* MDG1 growth was not negatively affected.

The LIVE/DEAD assay was performed to qualitatively evaluate the bacterial viability in planktonic and sessile phases. No significant differences in the bacterial green fluorescence viability were observed in the presence of all tested XYL and GOS concentrations for all tested strains in planktonic and sessile phases, confirming a bacteriostatic effect of XYL and GOS. Similar results were obtained for each detected skin emerging prebiotic combination. These results were also confirmed in the co-culture assay. Figure 5 shows representative LIVE/DEAD images of *S. aureus* 815, *S. aureus* PECHA 10, *S. epidermidis* 317 and *S. epidermidis* MDG1 biofilms after 24 h treated with the most performing prebiotic combinations. A general reduction of *S. aureus* strains cell aggregation was displayed in the presence of the tested combinations compared to the respective untreated samples without loss of bacterial viability. On the contrary, no differences in terms of clusterization were observed in *S. epidermidis* sessile growth compared to the respective untreated controls.

## 4. Discussion

Skin microbiota contributes to maintaining physiological skin acidity and prevents the colonization of transient pathogenic bacteria; moreover, the metabolites produced by the resident microorganisms suppress cutaneous oxidation [25]. The unbalance of commensal and pathogenic microorganisms is the cause of skin damage. Antibiotics and biocides are not the best choices to treat unbalanced skin microbiota [6]; a possible strategy is the use of prebiotics that, favoring the colonization of the autochthonous bacteria, protect the skin, achieving a dual action: a pathogenic bacteria reduction and a downregulation of their quorum-sensing-regulated gene expression [7]. As well known, prebiotics influence the host by the selective growth stimulation of beneficial bacteria improving its wellbeing. Moreover, FOS and GOS compounds are able to reduce the incidence of AD in high-risk newborns [25].

Therefore, the aim of this work was to evaluate the efficacy of prebiotic combinations on pathogenic *S. aureus* strains grown both in the planktonic and sessile mode without affecting the skin resident microbiota such as *S. epidermidis* strains.

The first step of the present study was to screen the best performing combination of XYL, an emerging skin prebiotic, and GOS, an established prebiotic, in terms of antimicrobial/antibiofilm species-specific actions against skin Staphylococcal strains. Among the all tested combinations, XYL and GOS combined at 1% concentration showed the most performing results in terms of selective species-specific antibacterial/antibiofilm action and low concentration. This combination can be considered able to balance the skin microbiota by blocking the pathogens colonization. For this reason, 1% XYL plus 1% GOS was chosen as the most performing since, at low concentrations, the selective species-specific antibacterial/antibiofilm action was extensively recognized. These data demonstrate the effective action of 1% XYL plus 1% GOS suggesting their use in future topic formulations.

Xylitol, Generally Recognized as Safe, expresses a significant antibiofilm effect in chronic wound infection and a relevant effect disrupting biofilm in the endodontic environment [15,26]. In addition, in cases of skin disorders, XYL displays several properties, including hydrating effects and microbial skin balance [27,28]. Katsuyama et al. [25] reported that the antibiofilm action of XYL, due to the suppression of the extracellular matrix production, was related to its low fermentability by *S. aureus*.

As well known, GOS is a compound contained in breast milk that protects newborns from infections without having bactericidal activity and not inducing bacterial resistance. Akiyama et al. [16] showed that 5% GOS inhibited the growth and the cell adhesion of *S. aureus* without hindering the *S. epidermidis* growth. Galacto-OligoSaccharides act by inhibiting *S. aureus* adhesion, and XYL blocks the *S. aureus* biofilm matrix production [16].

The second step of the present study was to identify more species-specific activities when XYL was combined with other skin emerging prebiotics aiming to propose novel associations aside from 1% XYL and 1% GOS.

A selective species-specific action against staphylococcal clinical strains was detected when XYL was combined with FOS (at 24 h) or IMO (at 3 h) or LAG (at 3 h), showing a major *S. aureus* growth reduction in respect to the reference control (1% XYL plus 1% GOS).

The species-specific action depended on the contact time and was, generally, more relevant after 3 h; only with FOS was recorded a stable effect during the time. Fujiwara et al. [29] showed in mice that FOS reduced the severity of AD-like skin lesions, decreasing the expression of IL-12p40 and the production of IL-12, IL-23-driven, IL-17 and IL-10.

Regarding the antibiofilm action, these prebiotic combinations (at the same times of planktonic phase) expressed a relevant reduction of *S. aureus* sessile population compared to *S. epidermidis* without interfering with microbial viability. No significant killing effect was detected.

When XYL combined with GOS or FOS or IMO or LAG were tested against *S. aureus* PECHA 10 and *S. epidermidis* MDG1 in co-culture, the most interesting selective action was detected after 3 h modulating the *S. aureus* PECHA 10 grown in respect to *S. epidermidis* MDG1. In particular, 1% XYL plus 1% IMO exerted a significant species-specific action with a marked presence of *S. epidermidis* MDG1 in respect to *S. aureus* PECHA 10. At 24 h, a balanced effect was obtained with 1% XYL plus 1% FOS.

In general, it can be concluded that the selected prebiotic combinations: (i) are able to modulate the Staphylococcal growth favoring the commensal microbial population in respect to the pathogenic one; (ii) express their best action in terms of selective species-specific effect after 3 h. Regarding this last consideration, it could be hypothesized that the microorganisms, after 24 h, are able to cope with an adaptive response to stressful prebiotics action. Hence, our data suggest a repeated administration for future plans of skin management, and the choice of one novel combination over another depends on the different potential applications.

The limitation of this study could be associated with the small number of clinical strains, although the *S. aureus* and *S. epidermidis* strains used for experiments displayed a similar general behavior within the species when treated with the different prebiotics alone or combined with each other.

Overall, it can be asserted that GOS, FOS, IMO and LAG combined with XYL at 1% concentration modulate the skin microbiota playing a role in the stabilization of indigenous beneficial strains and in the inhibition of pathogenic microorganisms. These combinations show selective species-specific action in planktonic and sessile phases of the studied strains and they are able to enhance the XYL prebiotic activity. In addition, in a dual Staphylococcal species mixed grown, the selected prebiotics combinations confirm their ability to restore the skin balance.

## 5. Conclusions

Despite the limitations indicated above, these data show the beneficial health effects of novel prebiotic combinations through the maintenance of skin microbial balance.

To the best of our knowledge, this is the first work in which different prebiotic combinations show a selective species-specific action on Staphylococcal strains.

All in all, our data suggest that the detected prebiotic compounds, combined in suitable concentrations, may have a role in keeping balance in skin microbiota by affecting a species-specific action between *S. aureus* and *S. epidermidis* strains. Furthermore, the recognized antibiofilm action underlines their role as potential antivirulence therapeutic agents.

Further investigations are needed to better understand the mechanisms of prebiotic action and to plan suitable formulations for topical skin care applications.

## Figures and Tables

**Figure 1 microorganisms-09-00037-f001:**
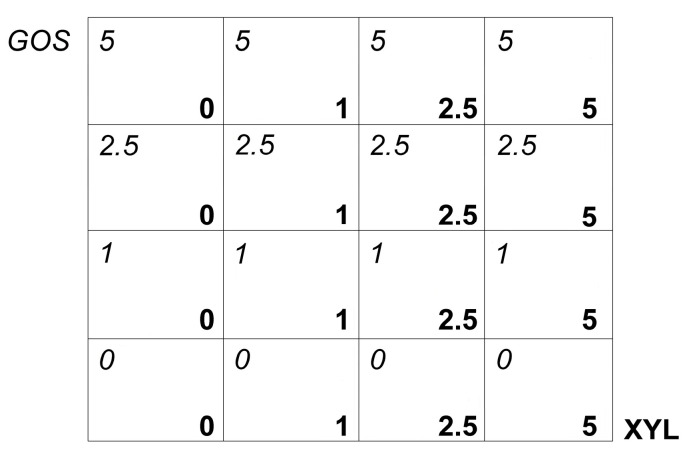
Scheme of Xylitol (XYL) and Galacto-OligoSaccharide (GOS) tested combinations.

**Figure 2 microorganisms-09-00037-f002:**
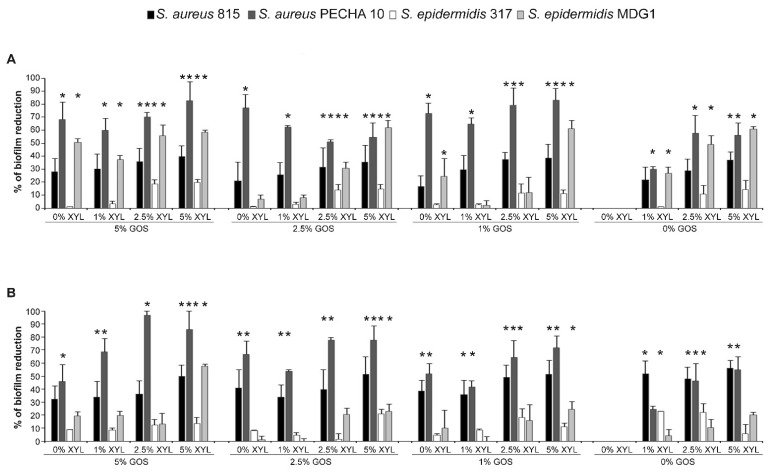
Percentage of reduction of *S. aureus* 815, *S. aureus* PECHA 10, *S. epidermidis* 317 and *S. epidermidis* MDG1 adhesion at 3 h (**A**) and biofilm formation at 24 h (**B**) after treatment with different combinations of Xylitol (XYL) and Galacto-OligoSaccharide (GOS) at different concentrations. * statistically significant with respect to the control.

**Figure 3 microorganisms-09-00037-f003:**
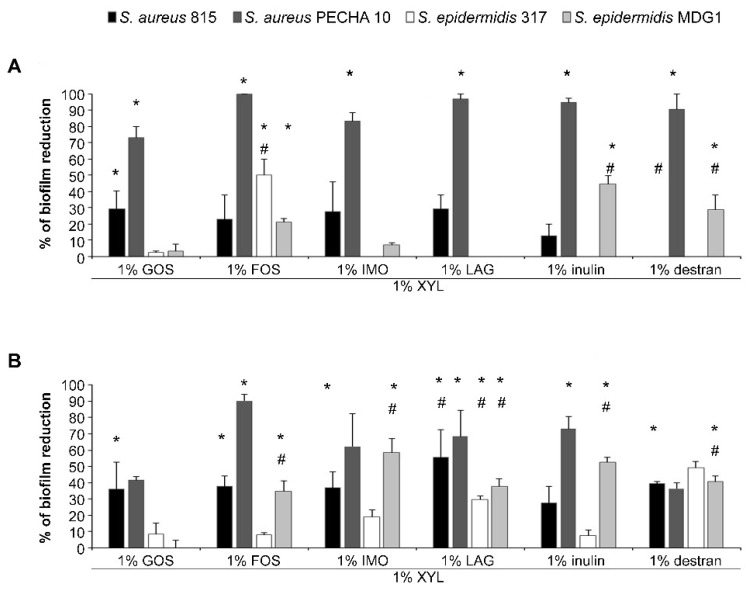
Percentage of reduction *S. aureus* 815, *S. aureus* PECHA 10, *S. epidermidis* 317 and *S. epidermidis* MDG1 adhesion and biofilm formation after treatment with 1% XYL and 1% FOS or IMO or LAG or inulin or dextran at 3 (**A**) and 24 h (**B**), respectively. 1% XYL and 1% GOS is the reference control (RC). For abbreviations, see materials and methods. * statistically significant with respect to the control. ^#^ statistically significant with respect to RC.

**Figure 4 microorganisms-09-00037-f004:**
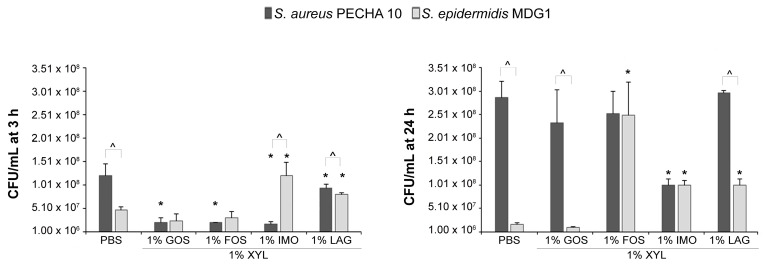
CFU/mL of *S. aureus* PECHA 10 and *S. epidermidis* MDG1 in co-culture assay after 3 and 24 h of prebiotics treatment. * statistically significant with respect to the corresponding strain in PBS. ^^^ statistically significant between the two tested strains. For abbreviations, see materials and methods.

**Figure 5 microorganisms-09-00037-f005:**
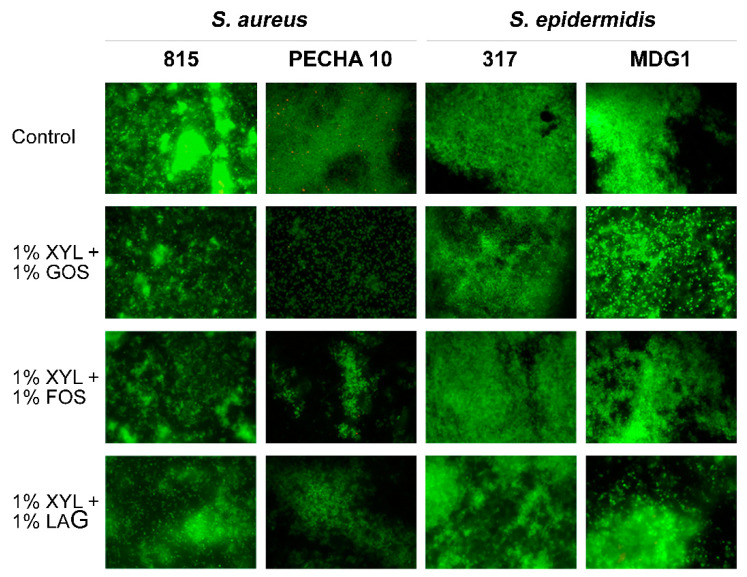
Representative LIVE/DEAD images of *S. aureus* 815, *S. aureus* PECHA 10, *S. epidermidis* 317 and *S. epidermidis* MDG1 biofilms treated with prebiotic combinations after 24 h. The prebiotic effect was markedly displayed comparing *S. aureus* strains to *S. epidermidis* strains biofilms. Sessile population in biofilms stained in red (propidium iodide) expressed compromised membrane integrity (damaged), whereas green stained bacteria (SYTO 9) remained viable. Original magnification 1000X.

**Table 1 microorganisms-09-00037-t001:** Percentages of planktonic *S. aureus* 815, *S. aureus* PECHA 10, *S. epidermidis* 317 and *S. epidermidis* MDG1 growth reduction (OD_600_) after 3, 6, 24 h of contact with Xylitol (XYL) and Galacto-OligoSaccharide (GOS) at different concentrations (1, 2.5, 5%).

**Strains**		**XYL%**											
***S. aureus*** **815**		**3 h**				**6 h**				**24 h**			
		**0**	**1**	**2.5**	**5**	**0**	**1**	**2.5**	**5**	**0**	**1**	**2.5**	**5**
**GOS%**	**5**	29.5 ± 15.4	34.26 ± 13.91	38.88 ± 12.18	40.6 ± 15.77	32.26 ± 10.40 *	41.8 ± 11.63 *	51.35 ± 6.67 *	55.50 ± 8.78 *	20.41 ± 13.39 *	17.5 ± 5.00 *	22.53 ± 7.27	21.83 ± 1.86 *
	**2.5**	18.7 ± 16.28	26.8 ± 15.66 *	32.24 ± 13.63 *	34.96 ± 15.53	22.35 ± 9.47	31.44 ± 11.62 *	37.39 ± 10.23	46.42 ± 13.11 *	9.06 ± 6.95	9.31 ± 4.81	15.67 ± 2.97	12.44 ± 9.72
	**1**	13.8 ± 13.79	22.4 ± 17.45	27.94 ± 16.14	30.23 ± 17.22	22.59 ± 9.78	34.54 ± 5.35	41.18 ± 8.92 *	43.60 ± 14.55 *	5.96 ± 5.30	11.12 ± 2.15	13.56 ± 3.22	12.50 ± 3.77
	**0**	-	14.38 ± 12.78	23.93 ± 10.65	26.70 ± 15.21	-	15.86 ± 7.17	27.8 ± 12.77	35.13 ± 13.48 *	-	5.47 ± 3.62	9.29 ± 2.43	7.82 ± 7.52
		**XYL%**											
***S. aureus*** **PECHA 10**		**3 h**				**6 h**				**24 h**			
		**0**	**1**	**2.5**	**5**	**0**	**1**	**2.5**	**5**	**0**	**1**	**2.5**	**5**
**GOS%**	**5**	51.24 ± 9.74 *	58.12 ± 1.11 *	59.78 ± 2.74 *	62.29 ± 2.95 *	21.23 ± 2.32	35.81 ± 0.85 *	44.83 ± 1.98 *	33.37 ± 3.71 *	12.88 ± 4.74	10.00 ± 14.14	18.46 ± 3.72	18.09 ± 1.35
	**2.5**	22.55 ± 3.03 *	50.90 ± 1.16 *	54.03 ± 0.62 *	58.93 ± 1.07 *	6.21 ± 8.78	11.29 ± 0.80	37.31 ± 2.81 *	24.47 ± 2.78 *	13.10± 18.52	4.78 ± 6.76	12.11 ± 17.12	17.09 ± 6.14
	**1**	4.07 ± 5.52	27.08 ± 9.21 *	43.55 ± 9.12 *	52.15 ± 0.06 *	1.21 ± 1.71	24.10 ± 0.14 *	3.29 ± 4.66	4.16 ± 5.88	0.00 ± 0.00	11.22 ± 15.86	12.91 ± 5.48	17.33 ± 2.61
	**0**	-	14.83 ± 3.33	19.73 ± 2.38 *	27.73 ± 0.11 *	-	1.20 ± 1.70	12.65 ± 17.02	3.13 ± 1.10	-	6.93 ± 9.80	12.99 ± 5.24	8.45 ± 4.55
		**XYL%**											
***S. epidermidis*** **317**		**3 h**				**6 h**				**24 h**			
		**0**	**1**	**2.5**	**5**	**0**	**1**	**2.5**	**5**	**0**	**1**	**2.5**	**5**
**GOS%**	**5**	31.3 ± 18.6	33.38 ± 17.00 *	38.70 ± 16.85	45.34 ± 14.10	36.67 ± 16.26 *	40.41 ± 10.95 *	45.70 ± 12.79 *	51.00 ± 69.59 *	12.51 ± 9.37	5.24 ± 4.53	11.34 ± 4.76	15.31 ± 14.88
	**2.5**	20.9 ± 20.50	32.16 ± 19.16	34.57 ± 16.10	38.81 ± 16.00	26.63 ± 15.27	34.75 ± 10.45	40.18 ± 13.29 *	45.87 ± 10.02 *	1.25 ± 1.08	5.65 ± 4.89	5.90 ± 1.84	32.62 ± 5.84
	**1**	10.8 ± 12.72	19.9 ± 22.40	27.10 ± 20.82	32.94 ± 19.28	15.92 ± 7.86	27.86 ± 14.24	37.78 ± 13.38 *	42.89 ± 10.60 *	6.03 ± 8.78	4.43 ± 4.02	12.04 ± 6.82	8.04 ± 3.48
	**0**	-	13.40 ± 18.63	19.3 ± 19.28	27.1 ± 19.08	-	22.11 ± 10.98	28.57 ± 11.14	29.99 ± 15.04	-	9.60 ± 3.22	8.60 ± 5.65	5.95 ± 5.23
		**XYL%**											
***S. epidermidis*** **MDG1**		**3 h**				**6 h**				**24 h**			
		**0**	**1**	**2.5**	**5**	**0**	**1**	**2.5**	**5**	**0**	**1**	**2.5**	**5**
**GOS%**	**5**	12.60 ± 16.88	19.52 ± 9.70	29.92 ± 3.66 *	39.35 ± 1.86 *	0.00 ± 0.00	13.88 ± 2.37	1.36 ± 1.92	5.14 ± 2.42	21.12± 7.46 *	24.43 ± 5.39 *	26.77 ± 4.18 *	32.78 ± 6.41 *
	**2.5**	0.00 ± 0.00	0.00 ± 0.00	5.52 ± 7.81	30.23 ± 4.10	0.00 ± 0.00	10.38 ± 7.66	9.94 ± 6.45	9.21 ± 5.63	13.28 ± 2.48	18.90 ± 3.67	12.22 ± 2.47	32.86 ± 3.30 *
	**1**	10.42 ± 4.34	1.33 ± 1.89	9.01 ± 2.34	24.19 ± 3.10	1.24 ± 1.76	6.43 ± 10.98	0.00 ± 0.00	23.21 ± 0.45 *	5.42 ± 2.66	7.02 ± 1.68	20.88 ± 0.84	37.02 ± 0.67 *
	**0**	-	9.84 ± 8.71	23.64 ± 10.88	29.44 ± 1.73 *	-	29.16 ± 7.20*	15.11 ± 1.04	32.82 ± 6.90*	-	4.87 ± 2.02	3.18 ± 4.50	22.74 ± 11.78 *

* Statistically significant with respect to the control. For the best comprehension, see the scheme in Figure 1.

**Table 2 microorganisms-09-00037-t002:** Percentage of planktonic *S. aureus* 815, *S. aureus* PECHA 10, *S. epidermidis* 317 and *S. epidermidis* MDG1 growth reduction (OD_600_) after 3, 24 h of contact with 1% of Xylitol (XYL) and 1% of new formulations Fructo-OligoSaccharides (FOS) or IsoMalto-Oligosaccharides (IMO) or ArabinoGaLactan (LAG) or inulin or dextran. 1% XYL and 1% GOS is the reference control (RC).

**Combinations °**	***S. aureus*** **815**		***S. aureus*** **PECHA 10**	
	**3 h**	**24 h**	**3 h**	**24 h**
1% XYL + 1% GOS	22.47 ± 17.45 *	11.12 ± 2.15	27.08 ± 9.21	11.22 ± 15.86 *
1% XYL + 1% FOS	42.44 ± 12.63 *	49.17 ± 21.46 *^#^	12.66 ± 3.06	0.00 ± 0.00 ^#^
1% XYL + 1% IMO	41.28 ± 4.88 *	0.00 ± 0.00	11.59 ± 9.11	0.00 ± 0.00 ^#^
1% XYL + 1% LAG	32.57 ± 4.60 *	0.00 ± 0.00	38.21 ± 5.31 *	3.42 ± 1.47 ^#^
1% XYL + 1% inulin	42.63 ± 3.94 *	0.00 ± 0.00	7.65 ± 1.30	0.00 ± 0.00 ^#^
1% XYL + 1% dextran	0.00 ± 0.00 ^#^	44.26 ± 1.01 *^#^	0.00 ± 0.00	7.78 ± 3.17 ^#^
**Combinations °**	***S. epidermidis*** **317**		***S. epidermidis*** **MDG1**	
	**3 h**	**24 h**	**3 h**	**24 h**
1% XYL + 1% GOS	19.94 ± 22.40	4.43 ± 4.02	1.33 ± 1.89	7.02 ± 1.68
1% XYL + 1% FOS	24.94 ± 10.82 *^#^	2.15 ± 3.04	14.03 ± 0.87 *	3.36 ± 1.78
1% XYL + 1% IMO	0.00 ± 0.00	0.52 ± 0.90	3.79 ± 0.64	27.77 ± 10.89
1% XYL + 1% LAG	0.00 ± 0.46	5.67 ± 0.87 *^#^	0.00 ± 0.00	45.04 ± 6.44 *
1% XYL + 1% inulin	26.27 ± 11.57 *^#^	4.12 ± 2.25 *^#^	18.00 ± 4.73 *^#^	36.72 ± 17.28
1% XYL + 1% dextran	0.00 ± 0.00	0.00 ± 0.00	0.00 ± 0.00	17.36 ± 9.45

° For abbreviations, see materials and methods. * Statistically significant in respect to the control. ^#^ Statistically significant in respect to RC.

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
