# Peer review of "Prebiotic Combinations Effects on the Colonization of Staphylococcal Skin Strains"

_microorganisms, 2020, doi:10.3390/microorganisms9010037_

Round 1

Reviewer 1 Report

In the present work, De Ludovico and co-authors describe the use of prebiotic combinations to diminish the growth and biofilm formation ability of Staphylococcus aureus in an species-specific manner. With this aim, they compared the impact of different combinations of xylitol, a well-known molecule with prebiotic properties with different kinds of compounds at different concentrations on two S. aureus clinical strains as well as on two different Streptococcus epidermidis strains. They also analyzed the viability of cells exposed to the prebiotic cocktail by the Live/Dead staining method.

In order to correctly evaluate the potential species-specific inhibitory properties of assayed cocktails, the authors should check these effects in co-cultures assays mixing S. aureus and S. epidermidis strains (i.e. PECHA 10 and MDG1) in presence of the cocktail and analyze the impact in each population (by plating on selective media for example). This assay is essential to clearly support their conclusions.

In the Material and Methods section, the authors present some properties of used strains such as hemolytic activity or the presence of agr alleles among others, but it is not clear why these characteristics are important in the context of their study. Also, it is not clear which blanks they have used for growth and biofilm formation measures. Can you include this information? Did the authors take into consideration the blanks values in their OD as well as percentage calculation?

Even if it seems clear that there is an inhibitory effect against the two assayed strains of S. aureus, the cocktail of xylitol (XYL) and Galacto-Oligosaccharide (GOS) at 1% each seems to have an impact on the growth of S. epidermidis, at least for the strain 317 based on values of table 2 (contrary to the sentence on lines 229-230). Besides, when we analyze the raw values of biomass production (Figure S1), the differences in growth are less clear, mainly at 3 hours where the growth is not higher than 0,2 for all the strains except S. aureus PECHA 10. Are these differences in growth statistically significant when comparing the OD values? This comment can be applied to the rest of the supplementary figures. In the case of growth inhibition results, some points should be verified:

  • Based on the selected cocktail (XYL + GOS 1%), there are similar inhibition percentages for all strains except for S. epidermidis MDG1 after 3 and 6 hours of exposition, which means that the inhibition is not species-specific.
  • there is a general reduction in the percentage values for almost all strains (except S. epidermidis MDG1) after 24 hours. It is an interesting phenomenon that might be discussed.

Furthermore, the authors present an important standard deviation in the tables and figures, which makes complicate to clearly see the differences. How many times have been repeated each experiment and with how many replicates? It is important to know the sample size (n) to evaluate if the statistical test is pertinent or not, and this variability could be reduced if they increase the number of replicates per condition.

The authors also analyzed the impact of prebiotic cocktails in biofilm formation of each strain at 3 and 24h. From a methodological point of view, it is not correct to evaluate the biofilm formation after 3 hours due to, even if it is dependent on bacterial species, after this time-lapse, the bacteria are common in the initial attachment to the surface before aggregations and biofilm development. So, this data should be carefully analyzed, and never as a biofilm structure.

Minor comments:

  • Line 177: this sentence is not clear at all; please, develop.
  • Line 223: “Galacto-Oligosaccharide” instead of “Gluco-Oligosaccharide”
  • Line 242: “(data not shown)”, please, provide this information as supplementary data.
  • Line 353: “extensively” instead of “exstensively”.`
  • Line 380: “behavior” instead of “beaviour”.
  • Line 399: “Furthermore” instead of “Furtermore”.

Reviewer 2 Report

The authors have incorporated improvements into their manuscript. 

The graphical presentation of the data adds information but could still be improved. Since the authors want to show that the combinations of prebiotics have strain-specific effects, grouping the different strains together and show their growth side by side for each combination of prebiotic might aid this agenda. 

Regarding the live/dead staining: The authors now describe the experiment more clearly. However, a description of the evaluation criteria should be added to the M+M section.

Round 2

Reviewer 1 Report

In the current version, Lodovico and co-authors improved their manuscript and answered correctly to almost all my comments and suggestions. I thank the authors for taking into consideration my advice concerning the co-culture assay to clearly demonstrate the species-specific activity of prebiotic-cocktails. Even if the experiment seems to be well-designed, I think that the analysis of the data is not correct. In fact, in figure 4, the authors compared and showed the statistical differences between S. aureus and S. epidermidis exposed to the same condition (i.e PBS, XYL + GOS,…). However, in order to correctly demonstrates the species-specific activity, the authors should compare each experimental condition to the control (PBS) for each species. To demonstrate their hypothesis, a decrease in the CFU/mL should be observed for one of the species whereas the other one should not be negatively affected.

Once again, I have a doubt concerning the statistical analysis performed. The authors used a t Student test to evaluate the impact of each cocktail on each species growth or biofilm formation ability. Based on their different comparisons to the “control” or “reference control”, I think that an ANOVA could be more adapted for multiple statistical comparisons.

Also, concerning growth inhibition, there is a general reduction in the percentage values for almost all strains (except S. epidermidis MDG1) after 24 hours. It is an interesting phenomenon that might be discussed. Is it an adaptative response of each species to each stressful condition? Please, develop.

Concerning the phenotypical characterization of the strains, I think that there are no necessary since there are no discussed or important for the main message of their manuscript, apart from to the biofilm formation ability.

Minor comments:

  • Line 44: the full name of Staphylococcus aureus should be included since its first mention. Please, invert “ aureus” and “Staphylococcus aureus” in this line.
  • Line 83: please, change the term “suggest”. The goal of a study is not to suggest but research, explore, analyze, …
  • Line 152: you stated that equal CFU/mL were mixed but you provide optical density value. Please, correct this sentence.
  • The legend of Figure 2 should be developed to better explain all showed results.
  • Line 480: “co-culture” instead of “co-coltore”.
